# Cuticle Hydrocarbons Show Plastic Variation under Desiccation in Saline Aquatic Beetles

**DOI:** 10.3390/insects12040285

**Published:** 2021-03-25

**Authors:** María Botella-Cruz, Josefa Velasco, Andrés Millán, Stefan Hetz, Susana Pallarés

**Affiliations:** 1Departamento de Ecología e Hidrología, Universidad de Murcia, 30100 Murcia, Spain; jvelasco@um.es (J.V.); acmillan@um.es (A.M.); 2Department of Animal Physiology/Systems Neurobiology and Neural computation, Humboldt University, 10115 Berlin, Germany; skhetz@skhetz.de; 3Departamento de Biogeografía y Cambio Global, Museo Nacional de Ciencias Naturales, CSIC, 28006 Madrid, Spain; susana.pallares@um.es

**Keywords:** CHC profiles, cuticle permeability, desiccation, aridification, aquatic insects, physiological plasticity

## Abstract

**Simple Summary:**

Desiccation resistance and physiological plasticity are key traits for species persistence in the context of aridification under global change. One of the main mechanisms of desiccation resistance in insects is the control of cuticular transpiration through changes in the quantity and composition of epicuticular hydrocarbons (CHCs), which have been well studied in terrestrial insects. Our study provides novel information regarding the capacity to cope with desiccation stress through plastic changes in the composition of cuticle hydrocarbons in aquatic insects from saline intermittent streams. We demonstrate that the plasticity of CHCs is an effective mechanism to regulate water loss rate under desiccation stress in the most saline-tolerant species studied. These traits, so far largely unexplored in aquatic insects, are relevant to understanding different biochemical adaptations to deal with drought stress in inland saline waters in an evolutionary and ecological context.

**Abstract:**

In the context of aridification in Mediterranean regions, desiccation resistance and physiological plasticity will be key traits for the persistence of aquatic insects exposed to increasing desiccation stress. Control of cuticular transpiration through changes in the quantity and composition of epicuticular hydrocarbons (CHCs) is one of the main mechanisms of desiccation resistance in insects, but it remains largely unexplored in aquatic ones. We studied acclimation responses to desiccation in adults of two endemic water beetles from distant lineages living in Mediterranean intermittent saline streams: *Enochrus jesusarribasi* (Hydrophilidae) and *Nebrioporus baeticus* (Dytiscidae). Cuticular water loss and CHC composition were measured in specimens exposed to a prior non-lethal desiccation stress, allowed to recover and exposed to a subsequent desiccation treatment. *E. jesusarribasi* showed a beneficial acclimation response to desiccation: pre-desiccated individuals reduced cuticular water loss rate in a subsequent exposure by increasing the relative abundance of cuticular methyl-branched compounds, longer chain alkanes and branched alkanes. In contrast, *N. baeticus* lacked acclimation capacity for controlling water loss and therefore may have a lower physiological capacity to cope with increasing aridity. These results are relevant to understanding biochemical adaptations to drought stress in inland waters in an evolutionary and ecological context.

## 1. Introduction

Among fitness-related physiological traits, those associated with water balance are highly relevant for insects in natural habitats [1]. The success of insects in terrestrial environments is largely due to the waterproofing properties of their cuticles to resist desiccation stress, which has been the subject of many studies on terrestrial arthropods (e.g., [2,3,4,5]), but is relatively unexplored in aquatic insects that experience desiccating conditions during drought events. Drought is a critical stressor in many freshwater ecosystems [6,7]. In intermittent systems of arid and semiarid areas, flow connectivity is disrupted and some water bodies may remain completely dry for long periods during seasonal droughts. Adults of some aquatic insects, like water beetles, fly from drying sites to more favorable wet habitats, experiencing important water loss because of the exposure to air and the extra dehydration associated with flight activity [8,9,10]. Therefore, the desiccation resistance of the adult (dispersive stage) may constrain the dispersal and survival strategies of these species [11,12]. In Mediterranean regions, where dry events are becoming more intense, prolonged and unpredictable with ongoing climate change [13], desiccation resistance will be a key trait for the persistence of many aquatic species. However, our understanding of the specific mechanisms by which aquatic insects deal with desiccation stress is still very limited.

One of the main mechanisms of desiccation resistance in insects is the control of cuticular transpiration [14,15], which generally accounts for the greatest proportion of body water loss [16,17]. The variation of cuticular water loss is driven by the permeability of the epicuticular hydrocarbons (CHCs) [15,17,18]. In general, a higher amount of CHCs with longer carbon chains, higher linearity and saturation confer a better waterproofing capacity to the cuticle [4,19,20]. In some terrestrial insects, CHC profiles show relatively rapid plastic changes in composition to adjust to the current environment, for example in response to changing temperature or desiccation conditions, (e.g., [5,21,22]). The overall CHC amount can be also increased in response to drought stress (e.g., [22,23,24,25,26]). Such changes have been proven to confer increased desiccation resistance (e.g., [2,4,27,28,29]).

Beetles from intermittent saline waters represent an interesting study case to assess whether the CHC’s composition and plasticity also play a relevant role in the desiccation resistance in aquatic insects, for several reasons. First, salinity and desiccation are co-occurring stressors in their habitats, and both induce osmotic stress and alter water balance, producing cuticular water loss. Indeed, some species have shown cross-tolerance (as defined by [30]) to these stressors [31]. These habitats are also characterized by a high climatic and hydrological variability, which may select for the evolution of high physiological plasticity [32,33,34]. Accordingly, it has been demonstrated that acclimation to high salinity induces changes in CHC composition that reduce cuticle permeability in saline water beetles [35]. Therefore, it is likely that desiccation induces plastic changes in CHCs similar to those elicited by salinity, and the modulation of cuticular permeability could be one of the possible shared protective mechanisms underlying cross-tolerance to both stressors. Finally, it has been suggested that desiccation resistance provided the physiological basis for the development of osmoregulation ability in some water beetle lineages, enabling the colonization and diversification across meso- and hypersaline habitats [36,37]. Then, a better knowledge of the specific strategies by which saline aquatic beetles cope with desiccation stress might shed light on the mechanistic and evolutionary links between these correlated stressors and contribute to understanding the success of some particular insect lineages in highly stressful environments such as intermittent saline waters [38].

The objective of this study was to explore if saline beetle species representative of independent evolutionary lineages that colonized such naturally stressful habitats show plastic variation in cuticular permeability (cuticular water loss rate) and CHCs quantity and composition in response to desiccation stress. For that purpose, we focused on the Iberian endemic water beetle species *Enochrus jesusarribasi* Arribas and Millán, 2013 (Polyphaga, Hydrophilidae) and *Nebrioporus baeticus* (Schaum 1864) (Adephaga, Dytiscidae), common inhabitants of Mediterranean saline running waters [39]. Both species are effective osmoregulators, their basal tolerance to salinity and desiccation resistance have been well documented [31,40,41] and they have shown an extraordinary ability to rapidly adjust CHC composition in response to environmental salinity changes compared with freshwater congeneric species [35]. We predicted that these species, adapted to osmotic stress, will also show CHC plasticity in response to desiccation, and so, exposure to non-lethal desiccation conditions would enhance desiccation resistance under a subsequent desiccation stress. To test this hypothesis, we measured cuticular water loss rate and multiple CHC traits (proportions of different CHC classes, median chain lengths of each class and the whole CHC profiles). Under the hypothesized beneficial acclimation response, we would expect that prior desiccation exposure reduces cuticular permeability in a subsequent exposure and elicits changes in either CHC amount and/or composition associated with enhanced waterproofing capacity in insects (e.g., increase in total CHC amount, in the proportion of longer chain n-alkanes and branched compounds and decrease in the proportion of unsaturated ones).

## 2. Materials and Methods

### 2.1. Study Species, Field Collection and Maintenance

*Enochrus jesusarribasi* and *N. baeticus* are Iberian endemic species that inhabit intermittent saline streams in South Spain [39]. These species face frequent seasonal droughts in their localities by flying from drying to wet sites, and no other desiccation resistance strategies (e.g., diapause, desiccation-resistant eggs or larvae) are known [39]. Experimental studies have shown that *E. jesusarribasi* is more tolerant to osmotic stress (both salinity and desiccation) than *N. baeticus* [31,40,41], and they also differ in body size, morphology and microhabitat. *Nebrioporus baeticus* (6.4 ± 0.28 mg of fresh mass), with a hydrodynamic body shape, is a strong swimmer frequently found in the water column [42]. *E. jesusarribasi* is slightly larger (7.20 ± 0.64 mg of fresh mass), has a more round and convex body shape and is a “walker” species mostly associated with the shorelines [39].

Adult individuals of each species were collected in October 2016 from two intermittent saline streams located in Murcia (SE Spain): Rambla Salada (38°07’22.0” N, 1°06’39.5” W) (*E. jesusarribasi*) and Río Chícamo (38°12’43.7” N, 1°03’07.9” W) (*N. baeticus*). Electrical conductivity values, measured in situ with a conductivity-meter (HACH/Hq40d), were 84 and 20 mS cm^−1^, respectively.

In the laboratory, specimens were placed in aerated 4 L aquaria with saline water at the salinity conditions measured in the collection localities of each species (20 mS cm^−1^ for *N. baeticus* and 80 mS cm^−1^ for *E. jesusarribasi*). Such conductivities are within the optimal range for these species in nature [41] and were chosen to avoid additional osmotic stress previous to the experiments. Saline water was prepared by dissolving an appropriate quantity of marine salt (Ocean Fish, Prodac, Cittadella, Padua, Italy) in distilled water. They were kept at 20 °C and a 12:12 L:D cycle in an environmental chamber (SANYO MLR-351, Sanyo Electric Co., Ltd., Moriguchi City, Osaka, Japan) for 5 days prior to the experiments, for habituation to laboratory conditions. Food was provided daily (frozen chironomid larvae for *N. baeticus* and Ruppia sp. for *E. jesusarribasi*).

### 2.2. Desiccation Experimental Procedures

Groups of 50 individuals of each species were randomly assigned to either a water control group (C) or to a prior desiccation group (Figure 1). For this prior desiccation exposure, specimens were gently dried on blotting paper and then individually placed in open glass vials and exposed to 40 ± 5% RH and 20 °C for 6 h (*E. jesusarribasi*) or 3 h (*N. baeticus*) in the environmental chamber, at the constant temperature and light cycle conditions mentioned above (Figure 1). A relative humidity of 40% was selected because it represents the minimum air humidity in the natural habitats of these species. Temperature was set to 20 °C to avoid additional thermal stress in the experiment. Exposure times were set based on the differential survival times under desiccation shown by these species in a previous study [31], in which *N. baeticus* was more sensitive to desiccation than *E. jesusarribasi*. After prior desiccation, specimens were allowed to recover for 24 h by immersing them in water at their corresponding field optimum salinity. The control groups were simultaneously maintained in aquaria at the conditions described above. Then, specimens from both the control and prior desiccation groups were exposed simultaneously to subsequent desiccation of the same intensity and duration as the first desiccation treatment (40 ± 5% RH for 3 h in *N. baeticus* and 6h in *E. jesusarribasi*). These experimental groups are referred to as D1 (one desiccation phase) and D2 treatments (two desiccation phases), respectively (Figure 1). Survival was monitored after each experimental phase (i.e., prior desiccation, recovery underwater and subsequent desiccation).

### 2.3. CHC Extraction

To determine changes in CHC associated with desiccation, six individuals were randomly removed from the water control (C), and directly after desiccation (i.e., one desiccation in D1 treatment and the second desiccation in D2 treatment; see Figure 1), and immediately freeze-killed at −80 °C. CHC extraction, identification and quantification were made using gas chromatography–mass spectrometry (GC–MS) (see Botella-Cruz et al. 2017 [35], 2019 [43], for details on analytical procedures). The basic characterization of CHC structures was conducted by interpreting their EI mass spectra. N-alkanes were identified by comparison of retention times with n-alkane standards (C10-C40; Sigma Aldrich, St. Louis, MO, USA) and adjustments were made to peak time based on the time and area of the octadecane standard as indicated in [35] (p. 3) and [43] (p. 4). From each individual CHC sample, the following variables were calculated: (i) the total amount of CHCs, (ii) the relative abundance of each CHC as the proportion of its adjusted peak area on the total adjusted areas (i.e., the sum of all the CHCs in the corresponding sample), (iii) the proportion of the main CHC classes (n-alkanes, branched alkanes and unsaturated) and (iv) median chain length of each CHC class. After CHC extraction, adults were sexed by examining genitalia in a stereomicroscope (Leica M165C with a Leica MEB10 fiber optic illuminator).

### 2.4. Cuticular Water Loss

Individuals from D1 and D2 treatments were weighed before and after the desiccation exposure in an analytical balance (0.01 mg accuracy) to determine gravimetrically the rate of total water loss (WLR; mg h^−1^), i.e., the sum of cuticular and respiratory water loss (Figure 1).

The proportion of WLR accounted for by cuticular transpiration was estimated by measuring cuticular and respiratory water loss in an independent experiment. For this, we used the hyperoxic switch method by a non-invasive technique that modulates spiracular opening by gas composition [44].

Twenty adult specimens of each species, previously acclimated in the laboratory at their optimum salinity conditions (see above), were individually placed in empty, hermetically sealed 100 mL chambers coupled with microsensors (BME280, Bosch Sensortec, Germany) that monitor the air temperature, pressure, and relative humidity every 60s while movement was controlled by a IR motion sensor (VCNL4010, Vishay, Chester County, PA, USA). Sensors were connected via an 8 channel MUX to an Arduino board running custom written software. The chambers were placed in an aquarium with a thermostat to keep temperature controlled at 20 °C. Relative humidity was also kept constant at 40 ± 5% by continuous bubble humidification connected to each individual chamber. In addition, the aquarium was covered with aluminum foil to avoid stress by light and minimize specimens’ movement. In each measurement, an empty chamber was also used as a control. After reaching a steady state, we started measuring H_2_O vapor for 45 min at normoxic conditions (used as a control); later, we infused the chambers with pure O_2_ for 45 min, which caused a temporary decrease in spiracular area and a coincident, transient drop in CO_2_ output and H_2_O vapor loss. Those individuals which showed high activity during the trials were discarded for further analysis.

We calculated the respiratory WLR from the proportional decrease of H_2_O vapor under hyperoxic conditions and estimated cuticular WLR by subtracting respiratory WLR from total WL with the equations:Respiratory water loss (RWL): 100 (∆WL/WLR normoxia)
∆WL: WLR normoxia − WLR hyperoxia
Cuticle water loss (CWL): 100-RWL

Given the possibility that specimens did not maintain the spiracles fully closed during the complete hyperoxia period, we compared these estimates with those obtained from the first 15 min exposure. For simplicity, 45 min estimates were used because rates were similar.

The percentage of CWL estimated by this experiment was applied to the total WLR measured gravimetrically in the desiccation experiment.

### 2.5. Data Analysis

We compared the cuticular WLR during desiccation exposure between D1 and D2 treatments (see Figure 1). CHC traits and profiles were compared between both desiccation treatments plus the samples subtracted from the water control (C), as a reference of CHC characteristics in the absence of desiccation stress. The analyses were made for each species separately to avoid direct comparisons of WLR and CHC estimates in two distant species with different salinity optima and exposure times to desiccation. We compared instead the capacity of the species to display an acclimation response after prior desiccation exposure. Differences in water loss rates, total amount of CHCs, the relative abundances of the major CHC classes and median chain length of each class among treatments were assessed by analyses of variance (ANOVA) and Bonferroni post hoc tests, using R v.3.3.2 (R Development Core Team, 2019). Sex was initially included as a covariate and, for simplification, removed from further analyses because its effect was not significant. Normality and homoscedasticity assumptions were validated on model residuals by graphical inspection [45]. Variation in CHC profiles was investigated by a Partial-Least Squares Discriminant Analysis (PLS-DA). The multiple correlation coefficient (R2) and cross-validated R2 (Q2) were used to confirm the predictive power of the fitted models. The statistical significance of the PLS-DA was also assessed with permutation tests (1000 permutations). Variable Importance in Projection (VIP) scores, which are the weighted sums of the squares of the PLS loadings, were obtained by the PLS-DA to identify the hydrocarbons that contributed the most to the CHC profile differences (VIP > 1.5). The concentrations of CHCs were log-transformed. These analyses were conducted using the statistical software MetaboAnalyst 4.0 [46].

## 3. Results

### 3.1. Variation in Cuticular Water Loss Rates

The proportion of cuticular water loss over the total water loss at 40% RH represented 59% ± 3.78 (N = 6) in *E. jesusarribasi* and 90% ± 1.86 (N = 10) in *N. baeticus*. In *E. jesusarribasi*, prior desiccation induced a positive acclimation response: cuticular WLR in the D2 treatment measured after the second desiccation was 17% lower than that measured at the same time in the D1 treatment (i.e., individuals not previously desiccated) (*p* < 0.01, Figure 2). In *N. baeticus,* no significant differences between treatments were found (*p* > 0.05, Figure 2).

### 3.2. CHC Traits

In *E. jesusarribasi*, a total of 52 different compounds were identified, ranging from 40 to 42 among treatments (Table 1 and Appendix A). Hexatriacontane (n-C36) was the longest-chain CHC identified in this species, and the most abundant compound was an unsaturated one (n-C21) in all treatments.

Individuals from the D2 treatment showed significantly longer chain lengths of n-alkanes and branched alkanes than those from D1 and C treatments (Table 1). The greatest amount of CHCs was found in D1 samples (Figure 3a, Table 1). D2 individuals presented significantly higher proportion of branched alkanes and a lower relative abundance of unsaturated compounds than the D1 and C groups (Figure 3, Table 1).

In *N. baeticus*, 54 different compounds were identified, ranging from 41 to 48 among treatments (Table 1 and Appendix A). The CHC with the longest carbon chain was hentriacontane (n-C31), and the most abundant one was an unsaturated compound (n-C21) in the control and tricosane in both desiccation treatments. The median chain length of unsaturated compounds was slightly but significantly longer in individuals from D2 than D1 and C treatments (*p* < 0.05, Table 1). CHC amount differed significantly among all treatments, being the highest in the control and the lowest in D2 individuals (*p* < 0.05, Figure 3, Table 1). No significant differences were found in the relative abundances of any CHC class among treatments (Table 1, Figure 3b,d).

### 3.3. CHC Profiles

In both species, the PLS-DA revealed significant differences in CHC profiles among treatments (permutation tests; *p* < 0.05, 0/1000). The PLS–DA analysis returned the first (LD1) and second (LD2) linear discriminant axes, which accounted for 36.1% and 24%, and 26.1% and 19% of the total variance in *E. jesusarribasi* and *N. baeticus*, respectively (Figure 4a,b). In both species, CHC profiles of individuals of the C group were located on the negative side of the first axis (LD1) and those from D2 were on the positive side, with D1 individuals in an intermediate position.

In *E. jesusarribasi*, the VIP scores analyses showed that differences in CHC composition among treatments were principally due to the abundance increase of long-chain n- alkanes (hexatriacontane, triacontane) and branched alkanes (3-methyl-heptacosane, compound 67; Appendix A) along desiccation treatments, and the biosynthesis of new long branched alkanes (10-methyl-octacosane and compounds 59, 71; Appendix A) in the individuals exposed to D2 (Figure 4c). In *N. baeticus*, the abundance of 11/13-methyl-heptacosane was higher in individuals from D1, and nonacosane was only present in the individuals from D2. Other compounds that contributed to profile differences were n-C21 unsaturated (compounds 3 and 5) and a n-C29 branched alkane (compound 58; Appendix A), showing the maximum abundance in the control and being absent in those individuals from D2 desiccation treatment (Figure 4d).

## 4. Discussion

Our study shows that desiccation exposure induces changes in CHCs in two aquatic beetles living in intermittent saline streams from semi-arid regions that represent distant evolutionary lineages of aquatic Coleoptera. However, the chemical strategies to cope with desiccation and acclimation capacity differed between the studied species.

*E. jesusarribasi* showed a clear beneficial acclimation response, increasing its desiccation resistance after a prior non-lethal desiccation hardening, in agreement with its high tolerance to salinity and desiccation reported in previous studies [31,40,41]. Such acute response to desiccation stress was associated with a change in cuticular hydrocarbons, as has been observed in some terrestrial insects, like *Drosophila melanogaster* [4]. In *N. baeticus*, the acclimation response was not found, as WLR did not differ between treatments (pre-desiccated and not pre-desiccated individuals) in this species (Figure 2).

The differential variation pattern in CHC traits shown by the two species (Figure 3) is congruent with such contrasting acclimation response to desiccation stress. In *E. jesusarribasi*, individuals from D2 displayed some of the expected CHC changes associated with increasing waterproofing capacity in insects: a higher abundance of branched alkanes, higher median chain length of alkanes and branched alkanes, and a lower abundance of unsaturated compounds than those not subjected to prior desiccation. Likewise, the most common responses in terrestrial insects subject to desiccation are the increase of the proportion of n-alkanes and the reduction of unsaturated CHCs, as seen, for example in ants [5], mosquitoes [3] and flies, even after a few hours of exposure to desiccation [2,4]. Branched alkanes also have an essential role in the waterproofing properties of the cuticle. They are quite common in terrestrial insect species from dry areas [47] and the predominant class of CHCs in desert Tenebrionidae, with an exceptionally thick and impermeable epicuticular wax layer, and generally with more carbon atoms than n-alkanes [48]. In addition, a recent study showed that a gene related to the synthesis of methyl-branched hydrocarbons is a determinant for the waterproofing properties of *Rhodnius prolixus* cuticle under desiccation [49].

*E. jesusarribasi* also increased the total CHC amount under desiccation (but only in the D1 desiccation treatment) as found in response to acclimation at high salinity in this species [35] and also in different terrestrial insects exposed to desiccation stress (e.g., [22,23,24,25,50,51]). However, in the subsequent desiccation (D2), a reduction of the total amount of CHC was observed, possibly indicating the responses to cope with desiccation varies with exposure time and under repeated exposures. Increasing the amount of the available CHCs could be a more immediate response to rapidly reduce cuticular water loss, while under longer or repeated stress, the strategy might shift towards a more complex response involving the synthesis of new compounds.

In *N. baeticus*, the changes in CHC typically related to improved waterproofing capacity were not observed. This result was consistent with the inability of this species to reduce WLR when exposed to desiccation after a prior desiccation. In this species, the total amount of CHCs decreased (a pattern also observed in response to acclimation at high salinity in [35] (pp. 4–5)), and the median chain length of unsaturated hydrocarbon increased. Therefore, these changes may reflect a physical response of the CHC blend to dryness rather than a physiological mechanism to control water loss. Indeed, these CHC changes would reduce overall waterproofing, although some authors found that alkenes also increase with temperature [22,52].

Although they exhibited different patterns in quantitative terms, both species showed changes in CHC profiles associated with desiccation, which involved the increase of the abundance of some CHCs and the biosynthesis of new hydrocarbons (Figure 4). Adjustments in CHC composition that involve the synthesis of new saturated n-alkanes and longer-chain branched alkanes were also common in the acclimation and acclimatization responses to high-salinity conditions in these species [35]. As we found, the largest acclimation effects often concern strongly aggregating (e.g., n-alkanes) and strongly disruptive (e.g., multiply methyl-branched alkanes) compounds [26]. In the two studied species, salinity and desiccation induce changes in the cuticle, suggesting a pivotal role of this protective structure to deal with multiple stressors and a possible driver of cross-tolerance responses [31]. However, the specific changes in abundance of CHCs in response to salinity or desiccation are different [35], as well as their effectiveness in reducing cuticle permeability, which reflects the complexity of the studied responses and likely an important interplay between exposure time, stress type and intensity. This opens emerging interesting research questions to further explore the relationship between cuticle chemistry and waterproofing capacity in insects exposed in nature to different sources of osmotic stress, such as saline water beetles.

The differences in desiccation resistance between the studied species could reflect their different evolutionary histories [37,53,54]) and their different adaptations to desiccation to face current and future aridification in the context of climate change. Within the Coleoptera, the evolution pattern of genes related to CHC synthesis to face desiccation is more variable than in other groups of insects [55]. It is possible that the *Enochrus* species studied here belongs to a lineage with more recent terrestrial (and desiccation-resistant) ancestors than the *Nebrioporus* one, despite Hydroporinae (to which *Nebrioporus* genus belongs, see [56]) being a younger lineage than Hydrophilinae [53]. This hypothesis is supported by ancestral reconstruction analyses that suggest that desiccation resistance was ancestrally high in the subgenus *Lumetus* (including *E. jesusarribasi*) [37] and by the frequent secondary colonisations of the terrestrial medium (and back to water again) within the family Hydrophilidae [57,58,59], apparently not common within the suborder Adephaga [53,60]. Thus, *E. jesusarribasi*, could have a better physiological capacity to deal with increasing aridification and be less vulnerable to climate change than *N. baeticus*.

In an ecological context, the differences in desiccation resistance and physiological acclimation capacity between the studied species could be also related to different microhabitat occupations and behavioral plasticity. Both species occupy habitats characterized by a high daily and seasonal climatic variation, which is generally associated with high acclimation capacity [32,33,34]. However, such variability might differ at a microhabitat scale. Dytiscids as *Nebrioporus* species are strong swimmers, which might confer them a better ability for behavioral stress avoidance by accessing a wide range of microhabitats within the water column. Conversely, the hydrophilid *Enochrus* species are poor swimmers and mostly associated with the shorelines of aquatic environments [39], where environmental conditions are more extreme and fluctuating, and thus where they could be exposed to higher desiccation pressure. A limited potential for behavioral plasticity in *E. jesusarribasi* may have favored the evolution of greater plasticity in physiological traits [61], in consistency with the “Bogert effect” [62,63].

Further studies of desiccation resistance traits of saline and freshwater species within each genus, of well-known taxonomy and habitat preference, and accounting for phylogenetic relationships, would shed light on these questions. Besides, transcriptomic studies of the gene expression patterns associated with these CHC changes and their function would help to elucidate the evolutionary constraints on these traits in these water beetle lineages.

## 5. Concluding Remarks

We have demonstrated that insect cuticle composition and its plasticity play a fundamental and rather complex protective function against desiccation stress in saline aquatic insect species. Species representative of two different aquatic Coleoptera suborders (Adephaga and Polyphaga) showed changes in cuticular hydrocarbons composition and quantity from a relatively short exposure to desiccation, but the variation pattern differed between them. Prior desiccation in *E. jesusarribasi* induced a positive acclimation response through CHC changes similar to those found in response to salinity stress, mainly the increase in the relative abundance of methyl-branched compounds and higher chain length of alkanes and branched alkanes, showing that they were effective in decreasing cuticular permeability to avoid water loss. In *N. baeticus*, the observed CHC changes did not increase the waterproofing capacity, so their ecological function remains to be further explored. These interspecific differences reflect the importance of the evolutionary history and ecology in driving responses of species exposed to similar stressors (salinity and desiccation). Our results suggest that *E. jesusarribasi*, the species with higher tolerance to salinity and desiccation and acclimation capacity, could have a better physiological capacity to deal with increasing aridification and be less vulnerable to climate change than *N. baeticus*.

## Figures and Tables

**Figure 1 insects-12-00285-f001:**
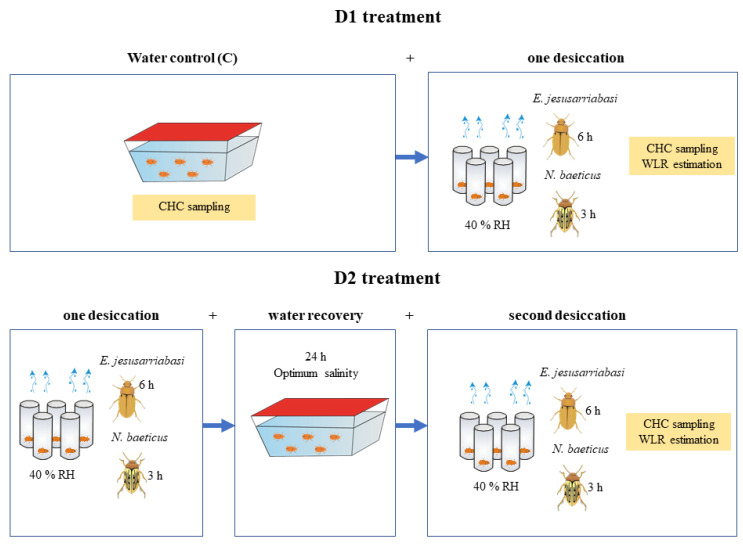
Schematic representation of the experimental design showing the different stress exposure steps and timings for cuticular hydrocarbons (CHCs) sampling and total water loss rate (WLR) estimation.

**Figure 2 insects-12-00285-f002:**
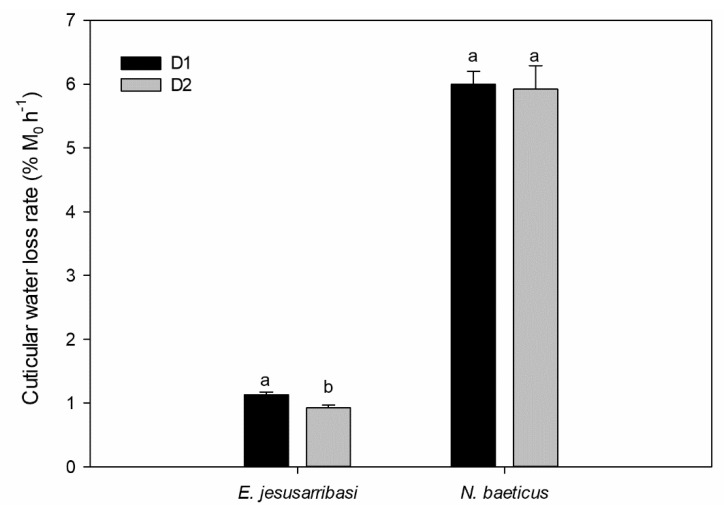
Cuticular water loss rate measured in D1 (individuals from the control exposed to desiccation) and D2 (individuals from the prior desiccation exposed to a subsequent desiccation). Lowercase letters indicate significant differences between treatments in Bonferroni post hoc tests (*p* < 0.05).

**Figure 3 insects-12-00285-f003:**
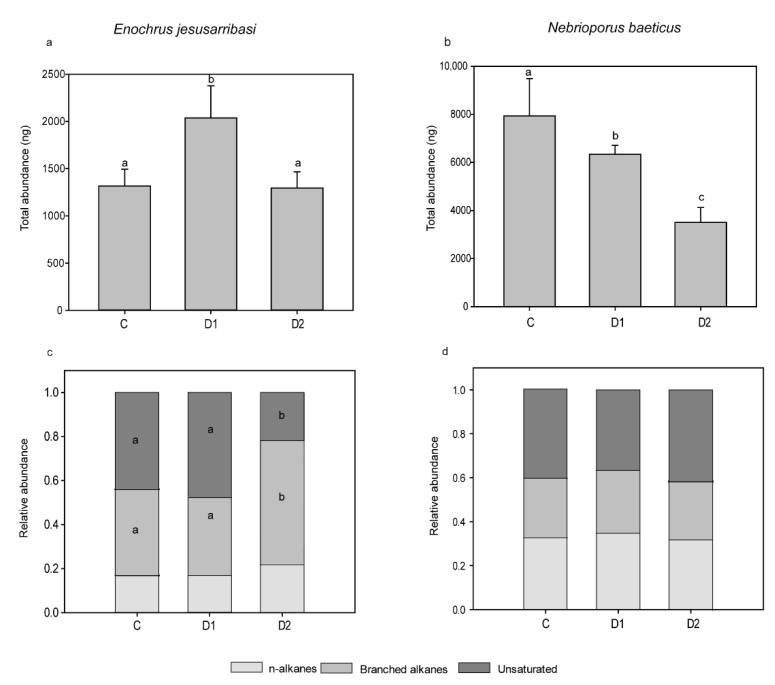
Total amount of cuticular hydrocarbons (CHCs) (Mean ± S.E.) (**a**,**b**), and mean relative abundance of the major CHC classes (**c**,**d**) in the different treatments (C: water control, D1: individuals from the control exposed to desiccation and D2: individuals from the prior desiccation exposed to a subsequent desiccation) of *Enochrus jesusarribasi* (**left**) and *Nebrioporus baeticus* (**right**). Lowercase letters indicate significant differences in Bonferroni post hoc tests (*p* < 0.05) between treatments (**a**,**b**) and between treatments within each major CHC class (**c**,**d**).

**Figure 4 insects-12-00285-f004:**
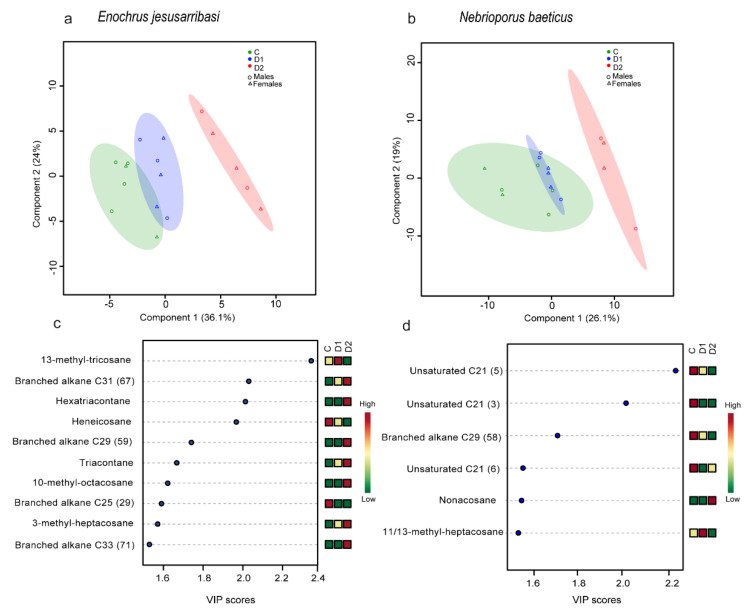
Score plots of the two principal components of cuticular hydrocarbons (CHC) concentration in *Enochrus jesusarribasi* (**a**) and *Nebrioporus baeticus* (**b**) (each dot represents an individual sample, ellipses represent the 95% confidence region for each species sample group, and the explained variances are shown in both axes in brackets). The variable importance plots (VIP) show the CHC compounds that contributed the most to the first axis based on their VIP scores for *E. jesusarribasi* (**c**) and *N. baeticus* (**d**) (colored squares represent the relative CHC concentrations of the corresponding compounds in each species; the corresponding ID number in Appendix A is indicated in brackets). C: water control, D1: individuals from the control exposed to desiccation, D2: individuals from the prior desiccation exposed to subsequent desiccation.

**Table 1 insects-12-00285-t001:** Total abundance of cuticular hydrocarbons (CHC), number of CHC compounds, median chain length (CL) (total and for each CHC main class) and number and proportion of compounds of each class (number of CHC of the corresponding class over the total) for the studied species. D1: individuals from the water control exposed to desiccation, D2: individuals from the prior desiccation exposed to a subsequent desiccation treatment.

Species	Treatment		Total n° CHC	Median CL ± SE	Alkanes						Unsaturated		
		Total Abundance (ng)			n-Alkanes			Branched Alkanes					
					n°	%	Median CL ± SE	n°	%	Median CL ± SE	n°	%	Median CL ± SE
*E. jesusarribasi*	Control	1315.92 ± 177.04 a	42	25.62 ± 0.09	8	16.91	25.96 ± 0.15 a	26	39.16 a	26.06 ± 0.06 a	8	43.93 a	23.83 ± 0.25
	D1	2036.90 ± 339.58 b	42	25.44 ± 0.16	9	17.12	26.23 ± 0.15 a	25	35.45 a	26.76 ± 0.25 a	9	47.43 a	23.33 ± 0.17
	D2	1294.13 ± 171.13 a	40	26.56 ± 0.20	7	21.52	27.97 ± 0.68 b	26	56.65 b	27.72 ± 0.11 b	7	21.83 b	23.98 ± 0.22
*N. baeticus*	Control	7936.46 ± 1548.11 a	48	24.70 ± 0.06	8	32.11	25.99 ±0.05	26	27.24	25.01 ± 0.12	14	40.65	23.21 ± 0.05 a
	D1	6333.31 ± 379.04 b	47	24.81 ± 0.03	8	37.91	25.96 ± 0.10	25	27.40	25.12 ± 0.09	12	34.69	23.35 ± 0.06 a
	D2	3505.50 ± 618.20 c	41	24.66 ± 0.16	10	32.34	25.04 ± 0.16	21	25.97	25.10 ± 0.37	10	41.67	23.85 ± 0.29 b

1Lowercase letters indicate significant differences (*p* < 0.05) in the relative abundance (%) and median chain length (CL) of the main cuticle hydrocarbons (CHC) classes between C, D1 and D2 groups within each studied species, from the ANOVA and post hoc tests results.

## Data Availability

All data generated or analyzed during this study are included in this published article.

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
