# Peer review of "Cuticle Hydrocarbons Show Plastic Variation under Desiccation in Saline Aquatic Beetles"

_insects, 2021, doi:10.3390/insects12040285_

Round 1

Reviewer 1 Report

This is a very interesting study describing acclimation of CHC profiles to desiccation stress in aquatic beetles. The paper is very well-written with only a couple of translation errors. The methods and results are well described. My only significant issue is with the time spent comparing the responses of the beetles. With only a two species comparison, and not using the same treatment for each species, it is impossible to make any conclusions about why they are different. I would focus instead on the fact that the responses seem to be different and how those responses might affect susceptibility to climate change.  

Author Response

> Thank you for this positive evaluation of our study.

Please note that the species comparison we have made in the discussion is based on their different ability to display an acclimation response to desiccation and on the extensive knowledge on their physiology and ecology, covered in previous studies. No direct comparisons on the specific response variables studied here are made because each species was exposed to specific desiccation conditions (according to their different tolerances) (L 214-115). We believe that the discussion of the differences in acclimation capacities shown by the species according to their different evolutive history, ecology and physiology responses actually enriches our article because these species represent two very distant evolutionary lineages that inhabit intermittent saline streams. Such information also constitutes the basis to explore their different susceptibility to climate change (L 387-388). Anyway, we have highlighted in the ms the need of further comprehensive multi-species comparisons inside each genus to address some of the ecological and evolutionary hypotheses that emerged from our study (L 404-410).

Reviewer 2 Report

„Cuticle hydrocarbons show plastic variation under desiccation in saline aquatic beetles“

This manuscript investigates the role of cuticle hydrocarbons to desiccation resistance/water preservation in two aquatic beetle species, specifically the chemical identity of these CHCs, and their dynamic regulation under controlled regimes of repeated desiccation. This role of CHCs addresses an important topic in insect ecophysiology. Repeated regimes of desiccation allow to analyse for plastic responses to animals not previously exposed to desiccation. While one species studied here showed changes in the CHC profiles following desiccation associated with reduced water loss, the other species lacked such a regulatory control. These results are placed into the context of evolutionary adaptations and ecological conditions of microhabitats.

The study is carefully designed, well documented, and presented in a clear way. Results are relevant to the understanding of insect’s physiological adaptations in a dynamic environment, and the work fits well into the scope of this journal. The evolutionary aspects would be better served by a comprehensive multi-species comparison, which is however mentioned in the discussion. For a few points, the manuscript needs to address some shortcomings during revisions:

- Data are missing in Table 1 for N. baeticus from the D2 experimental group. As indicated in the explanation beneath the table for the “D2 groups within each studied species”, these data should be provided.

- As a consequence of this data omission, the following statement that median chain length are significantly increased in D2 animals (l. 272) is shown only for E. jesusarribasi. Include specifics of statistical analysis in the text.

- The names of species and genus are often not set in italics, please check on this (see l. 235, 236, 239, 247, 268, 287, 291, 296, etc).

- In the discussion, reference to the specific figures of the results that are explained would be helpful (l. 311 – 321).

l. 374 ...could reflect their different evolutionary histories – which possibly constrain/limit the range of CHC responses?

l. 387 what do you mean by a phylogenetically controlled context? Are there obvious candidate taxa fur siuch studies, where both the phylogenetic relationships, habitat use, and the diversity of species are reliably known?

l. 396 While this explanation is intuitively appealing, is there any evidence available for such habitat choice in N. baeticus?

l. 161 consider: (see Botella-Cruz et al. 2017 [35], 2019 [42] for details on analytical procedures)

Author Response

>We are very grateful for the positive comments on our work and for your useful suggestions. The following points have been addressed in the new version of the manuscript:

- Data are missing in Table 1 for N. baeticus from the D2 experimental group. As indicated in the explanation beneath the table for the “D2 groups within each studied species”, these data should be provided.

- As a consequence of this data omission, the following statement that median chain length are significantly increased in D2 animals (l. 272) is shown only for E. jesusarribasi. Include specifics of statistical analysis in the text.

> We have included the missing data for N. baeticus from the D2 experimental group in Table 1. Additionally, we have made reference to the specific results of statistical analysis in the “Results” section (L 272 and 273).

- The names of species and genus are often not set in italics, please check on this (see l. 235, 236, 239, 247, 268, 287, 291, 296, etc).

> In the new version, all the names of species and genera have been italicized.

- In the discussion, reference to the specific figures of the results that are explained would be helpful (l. 311 – 321).

> Thank you for this suggestion. In order to improve the explanation, we have included the references to figures in discussion (L 323 and L361).

-374 ...could reflect their different evolutionary histories – which possibly constrain/limit the range of CHC responses?

> In this sentence we refer to the different timeframe of the transition from terrestrial to aquatic environments and diversification in saline habitats between the two beetle families studied here, which may have influenced the evolution of resistance mechanisms to desiccation. The Enochrus species studied here belongs to a lineage with more recent terrestrial (and likely more desiccation resistant) ancestors than the Nebrioporus one (L 374-382).

-387 what do you mean by a phylogenetically controlled context? Are there obvious candidate taxa fur siuch studies, where both the phylogenetic relationships, habitat use, and the diversity of species are reliably known?

> We appreciate your comment. Species of both genera have a well know taxonomy, phylogeny and ecology (saline or freshwater habitats). So, they could be considered as good candidates for this kind of studies. Thus, in the text we referred to the need to study both, saline and freshwater species within each genus. Anyway, we have clarified this issue in the text (L 404-410).

-396 While this explanation is intuitively appealing, is there any evidence available for such habitat choice in N. baeticus?

> This is based on the extensive knowledge on the ecology of dytiscids (e.g. see Yee 2014 and Millán et al 2014 – we have included this references in the ms now) and our personal field observations.

-Yee, D. A., & Yee, D. A. (Eds.). Ecology, systematics, and the natural history of predaceous diving beetles (Coleoptera: Dytiscidae). 2014- Dordrecht: Springer Netherlands.

-Millán, A.; Sánchez-Fernández, D.; Abellán, P.; Picazo, F.; Carbonell, J.A.; Lobo, J.M.; Ribera, I. Atlas de los coleópteros acuáticos de España peninsular; Madrid: Ministerio de Agricultura, Alimentación y Medio Ambiente, 2014; ISBN 9788449114182.

-161 consider: (see Botella-Cruz et al. 2017 [35], 2019 [42] for details on analytical procedures)

> We have made the change as suggested.

Reviewer 3 Report

I previously reviewed this manuscript for a different journal. Unlike many (most?) manuscripts that are sent to a different journal, the authors have actually listened to the initial reviews and improved the manuscript accordingly. My previous comments have been dealt with well, so I have only a few minor comments:

L 239. Change ‘any’ to ‘no’

L 353. I suppose some evaporation of the shortest CHC could occur, but these are still fairly long molecules and would have a very low vapor pressure.

L 413. Change ‘increased’ to ‘increase’

Table 1. Where are the D2 data for N. baeticus?

Author Response

> We are thankful for all the previous and new suggestions made for you that for sure they have helped us improve the manuscript. All of the changes suggested in the attached PDF file have been corrected too in this new version of the manuscript.

L 239. Change ‘any’ to ‘no’

> Changed as requested.

L 353. I suppose some evaporation of the shortest CHC could occur, but these are still fairly long molecules and would have a very low vapor pressure.

> We agree that it may be a bit speculative to assume that the shortest alkenes here studied (>20 C atoms) have been evaporated at the temperature (20ºC) we made our experiment. To avoid confusion, we have removed that sentence.

L 413. Change ‘increased’ to ‘increase’

> Corrected as requested.

Table 1. Where are the D2 data for N. baeticus?

> We have included the missing data for N. baeticus from the D2 experimental group in Table 1.

Reviewer 4 Report

Very well-written, thoughtful, and interesting paper. Only minor edits, see attached. Would suggest adding the GPS coordinates of the collection sites. It would also be useful to explain why the conditions of 45% RH and 20C were selected for the pre-and post-desiccation studies. The cuticular water loss rate was expressed as % M/h. Shouldn't this be expressed in some type of surface area-specific measure? Could the authors indicate the cuticular permeability of these species (i.e., micro grams of water lost per square centimeter per hour per mmHg or Torr or KPa)? This way your results could be compared with much of what is in the literature.

Author Response

> We are thankful for the positive and constructive revision made. All changes attached by reviewer in the PDF file have been corrected in the new version of the manuscript.

> Following your suggestion, in the “Methods” section, we have specified the GPS coordinates of the collection sites (L 121- 122)

> As suggested, in the “Methods” section, we have justified the conditions selected for our desiccation study (L141-143). These conditions were established based on other studies referred in the text (Pallarés et al 2017). Temperature at 20ºC was chosen to avoid additional stress in the experiments and the 40% RH was selected because it represents the typical minimum air humidity in the natural habitat of these species. In these conditions, both species have shown high survival at this humidity during the exposure times employed here in previous studies (Pallarés et al., 2017).

- Pallarés, S.; Botella-Cruz, M.; Arribas, P.; Millán, A.; Velasco, J. Aquatic insects in a multistress environment: cross-tolerance to salinity and desiccation. J. Exp. Biol. 2017, 220, 1277–1286, doi:10.1242/jeb.152108.

> Regarding cuticular water loss rate, in previous pilot studies, we made an attempt to measure the surface area of these species but we were not able to get accurate measurements. To avoid an additional source of variability in our data, we decided to use the mass instead, which is also a typical way to express water loss rates (Harrison et al. 2012).

Harrison, J.F., Woods, H.A. and Roberts, S.P. 2012. Ecological and Environmental Physiology of Insects. Oxford University Press, London.